# Type 2 Diabetes Mellitus Coincident with Clinical and Subclinical Thyroid Dysfunctions Results in Dysregulation of Circulating Chemerin, Resistin and Visfatin

**DOI:** 10.3390/biomedicines11020346

**Published:** 2023-01-25

**Authors:** Mohammad Reza Tabandeh, Amal Sattar Taha, Hanaa Addai Ali, Mohammad Razijalali, Narges Mohammadtaghvaei

**Affiliations:** 1Department of Basic Sciences, Division of Biochemistry and Molecular Biology, Faculty of Veterinary Medicine, Shahid Chamran University of Ahvaz, Ahvaz 6135783151, Iran; 2Department of Chemistry, Faculty of Science, University of Kufa, Najaf 54001, Iraq; 3Department of Clinical Sciences, Faculty of Veterinary Medicine, Shahid Chamran University of Ahvaz, Ahvaz 6135783151, Iran; 4Department of Laboratory Sciences, School of Paramedical Sciences, Hyperlipidemia Research Center, Ahvaz Jundishapur University of Medical Sciences, Ahvaz 6135783151, Iran

**Keywords:** diabetes mellitus, thyroid dysfunctions, adipocytokines, inflammatory cytokines

## Abstract

The alterations of circulating adipocytokines have been reported in thyroid diseases or type 2 diabetes mellitus (T2DM), but such data in T2DM coincident with clinical and subclinical thyroid-dysfunctions are limited, and remain to be investigated. We studied the changes in serum chemerin, resisitin and visfatin in T2DM patients with thyroid dysfunctions, and their association with inflammatory and insulin resistance-markers. A total of 272 female and male Iranian participants were selected and divided into six groups: the euthyroid group, T2DM, T2DM coincident with clinical and sub clinical hypothyroidism (SC-HO, and C-HO), and T2DM coincident with clinical and sub clinical hyperthyroidism (SC-HR, C-HR).Demographic characteristics, serum levels of adipocytokines, thyroid hormones, inflammatory factors (IL1-β, IL-6 and CRP) and insulin resistance-markers were determined in all participants. T2DM patients with clinical thyroid dysfunctions showed higher levels of circulating resistin, visfatin, chemerin and inflammatory factors, compared with the T2DM group and T2DM coexisted with subclinical thyroid diseases. No significant differences were observed in circulating adipocytokines and inflammatory markers between T2DM coexisting with subclinical thyroid diseases and those without thyroid dysfunctions. Our results revealed that clinical thyroid dysfunction in T2DM patients was associated with elevated levels of circulating resistin, chemerin, visfatin and inflammatory factors, while no such alteration was detected in T2DM coincident with subclinical thyroid dysfunction.

## 1. Introduction

Thyroid diseases and type 2 diabetes mellitus (T2DM) are the two most common metabolic disorders that tend to coexist in patients [1,2]. It has been reported that 7.7% of the Iranian adult population aged 25–64 years have diabetes, among whom one-half are undiagnosed. An additional 16.8% of Iranian adults have impaired fasting glucose [3]. The prevalence of overt and subclinical hypothyroidism in patients with T2DM in Iran is between 8 and 20%. The prevalence of overt and subclinical hyperthyroidism in the general population with T2DM in Iran has been reported as being between 1 and 2%. Thyroid dysfunctions in patients with T2DM are more common in women than in men in Iran [4].

The relationship between thyroid disorders and diabetes mellitus is characterized by a complex interdependent interaction. Hyperthyroidism is associated with impaired glucose tolerance and hyperglycemia, due to elevated glucose absorption through the gastrointestinal tract, increased hepatic glucose output, enhanced adipose tissue lipolysis and reduced insulin turnover. On the other hand, hypothyroidism interferes with the action and metabolism of insulin, and induces insulin resistance. Several previous studies have shown altered thyroid hormones in patients with T2DM, especially those with poor glycemic control [5,6].

Recent findings have shown that thyroid hormones may influence whole-body metabolism by regulating the adipose-tissue secretory functions. The adipose tissue dysfunctions that are observed in diabetic patients alter the whole-body metabolism by affecting thyroid functions [7,8]. Adipocytokines are biologically active substances produced by adipocyte with different physiological functions. Abnormal secretion of adipocytokines has been reported in obesity, T2DM and thyroid dysfunctions [6,7,8]. Thyroid-hormone changes in patients or experimental animals with clinical thyroid dysfunctions have been reported to alter the secretion of some adipokines such as leptin, adiponectin, omentin, resistin, visfatin and chemerin [9,10,11,12]. Prior studies have shown that thyroid-stimulating hormone (TSH) receptor is expressed in adipocytes and that TSH regulates the secretion of adipocytokines from adipocytes by signaling to the TSH receptor (TSHR) [13]. Diabetes is also associated with the abnormal pattern of expression or secretion of adipocytokines from adipose tissue. The alterations of serum adipocytokines in clinical and experimental thyroid-dysfunctions and diabetes are associated with a higher risk of inflammation [8,14]. However there is a lack of concordance in the literature concerning differences in plasma or serum adipocytokines-concentrations in diabetic patients coexisting with thyroid dysfunctions [7,8].

Chemerin, resistin and visfatin are recently identified adipose-derived adipocytokines. Changes in serum concentrations of these adipocytokines have been previously reported in diabetes or thyroid dysfunctions [10,11,14]. Chemerin acts as a proinflammatory adipocytokine by stimulation of the cellular production of inflammatory cytokines, including TNF-α, and IL-6 [15]. The serum chemerin-level has been found to be increased in hypothyroidism and decreased in the experimental hyperthyroidism rat-model [11]. Alshaikh et al. [10] reported that serum-chemerin levels are higher in patients with hyperthyroidism, compared to the controls. Visfatin is another adipokine that was previously identified as a growth factor for early B-lymphocytes, and named as a pre-B cell colony-enhancing factor [16,17]. There is a considerable discrepancy in the literature concerning the alteration of visfatin concentration in thyroid dysfunctions [17,18,19]. Some researchers reported a higher visfatin concentration in hyperthyroid patients [17,18], while others report a lower concentration in hyperthyroidism [17]. Resistin is an adipose-derived cysteine-rich adipocytokine that acts as an important molecular link between obesity and insulin resistance [20]. There are limited and conflicting data concerning the serum level of resistin in hypothyroid patients [21]. The first human study indicated the lower circulating levels of resistin in hyperthyroidism patients in comparison with euthyroid subjects [22], while high serum-values of resistin were observed in hyperthyroid patients in another study performed by Yaturu et al. [23].

Contrary to clinical hypo-and hyperthyroidism, which have noticeable symptoms and cause thyroid-hormone disturbances, patients with subclinical thyroid dysfunctions (SCTDs) lack the signs and symptoms of overt thyroid dysfunction. SCTD is defined as serum free-T_4_ and free-T_3_ levels within their respective reference ranges, in the presence of abnormal serum-TSH levels [24]. The prevalence of SCTDs has been reported to be between 1 and 10% of adult population samples. SCTD is associated with an abnormal lipid profile, cardiovascular dysfunctions, systemic inflammation, and atherosclerosis risk [25]. For these reasons, attention has recently been focused on the possible relationship between adipocytokines functions and SCTDs.

Although alterations of serum adipocytokines in thyroid dysfunctions and diabetes mellitus have been previously reported, available data related to chemerin, resistin and visfatin in T2DM coexisting with thyroid dysfunctions, are very limited or have not been studied. Thus, the aim of this study is to investigate the alterations of serum chemerin, resistin and visfatin in patients with T2DM coexisting with thyroid dysfunctions. We also examine the associations between serum adipocytoknes and thyroid profile values, insulin resistance-markers and proinflammatory cytokines in diabetic patients with SCTDs.

## 2. Materials and Methods

### 2.1. Ethics Statement

All study participants were studied as part of a clinical protocol approved by the research ethics committee of Shahid Chamran University of Ahvaz, Ahvaz, Iran (EE/99.3.02.65854/scu.ac.ir). and informed written consent was obtained from all the study participants.

### 2.2. Subjects and Study Design

This was a cross-sectional study of alterations of some adipocytokines in T2DMs coincident with clinical and subclinical thyroid dysfunctions. A total of 272 patients (127 male and 145 female) were included in the study, with an average age of 46.4 ± 12.6 years. Patients with T2DM coexisting with thyroid dysfunctions that regularly attended the Iran medical laboratory (Ahvaz, Iran) between February 2016 and September 2020 participated in the study. The patients’ general information, such as sex, age, weight, height and body mass index (BMI) was collected after obtaining a detailed medical history and performing relevant examinations. According to the results of their diabetes status, T2DM patients were further divided into five groups: euthyroidism patients (*n* = 64), subclinical hypothyroidism (SC-HO; *n* = 41), clinical hypothyroidism (C-HO; *n* = 43), subclinical hyperthyroidism (SC-HR; *n* = 36), and clinical hyperthyroidism (C-HR; *n* = 33). Thyroid dysfunctions was classified as the following: clinical hypothyroidism (C-HO; TSH > 4.20 μUI/mL and FT4 < 0.93 ng/dL), subclinical hypothyroidism (SC-HO; TSH > 4.20 μUI/mL and FT4: 0.93–1.7 ng/dL), subclinical hyperthyroidism (SC-HR; TSH < 0.27 μUI/mL and FT4: 0.93–1.7 ng/dL) and clinical hyperthyroidism (C-HR; TSH < 0.27 μUI/mL and FT4 > 1.7 ng/dL) [19]. The control group comprised 55 healthy people adjusted for age, sex, and BMI with normal thyroid function and negative thyroid antibodies. The BMI was calculated as the body weight in kg/height^2^ (m^2^) (kg/m^2^).

The diagnosis of T2DM was performed in accordance with the 2014 American Diabetes Association diagnostic criteria for diabetes, including symptoms of diabetes mellitus (polydipsia, polyuria, polyphagia), random blood glucose level ≥ 11.1 mmol/L, fasting plasma glucose (FPG) ≥ 7.0 mmol/L, 2-h post prandial glucose; PPG ≥ 11.1 mmol/L and glycated hemoglobin (HbA_1c_) > 6.5% [26]. The patients undergoing anti-inflammatory and antioxidant therapy, or suffering from severe peripheral vascular disease, hematologic disorders, cancers, infectious diseases, liver or kidney failure, or cardiac insufficiency were excluded from the study. In addition, individuals with any illness leading to a lack of ability to complete the assessment were not included as subjects.

### 2.3. Blood Sample Collection

Blood samples were obtained from the median cubital vein in all subjects between 7 a.m. and 8 a.m. while seated, after 12 h of fasting and 30 min of rest. The serum was centrifuged at 4000× rpm at, 25 °C for 10 min, and then placed in a −80 °C freezer, for subsequent analysis.

### 2.4. Biochemical Analysis

The free T_3_ (fT3), free T_4_ (fT4), TSH and insulin concentrations were determined using AccuBind ELISA kits, as recommended by the manufacturer (MonoBind Inc., Lake Forest, CA, USA). Glucose was determined using the glucose oxidase method (ParsAzmoon, Tehran, Iran). The HbA1c level was measured using a latex agglutination immunoassay kit (Man Co., Ltd., Tehran, Iran) and an automatic analyzer (BT4500, Biotechnica Instruments, Roma, Italy). Serum TPOAb was measured with Rapid ELISA (Monobind, Lake Forest, CA, USA). The intra-assay and inter-assay CVs were 6 and 4%, respectively. An anti-TPO concentration of more than 9 IU/mL was considered as positive [27].

### 2.5. Adipocytokines and Proinflammatory Cytokine Measurements

An ELISA was performed to determine the serum concentrations of visfatin, resistin and chemerin (Biovision, Inc., Milpitas, CA, USA), as recommended by the manufacturers. The IL1-β, IL-6 and CRP levels were determined using available human ELISA kits (Biovision, Inc., Milpitas, CA, USA).

### 2.6. Assessment of Insulin Resistance

The Homeostatic Model Assessment for Insulin Resistance (HOMA-IR) was calculated using the following formula: fasting plasma glucose (mmol/L) × fasting serum insulin (mU/L)/22.5.

### 2.7. Statistical Analysis

Statistical analysis was performed using the SPSS Version 26.0 (IBM, Armonk, NY, USA) software. All data were presented as mean  ±  standard deviation (SD). The normality of data or equality of error variances were determined using the Shapiro–Wilk or Levene’s tests. Between-group comparisons were performed using one-way analysis of variance and a least-significant-difference test as the post hoc test. A univariate analysis-of-covariance using a general linear model was used to adjust for the effect of sex and BMI on changes in studied serum factors. Multivariate relationships between serum adipocytokine levels and other factors were assessed using linear-regression analysis in hypothyroid and hyperthyroid patients, separately. In a two-interaction-analysis regression model, each adipocytokine was assigned as a dependent variable, each factor was assigned as a predictor, and sex and BMI were assigned as covariates. Tests were considered to be statistically significant if the 𝑝-value was lower than 0.05. The *, **, *** and **** symbols were used to indicate significant differences between all patient groups with healthy populations at *p* < 0.05, *p* < 0.01, *p* < 0.001 and *p* < 0.0001, respectively.

## 3. Results

### 3.1. Demographic Information of Studied Population

There was no statistical difference in the age of patients among the six different groups (Table 1). The BMI of the T2DM and T2DM with hypothyroidism groups was higher than that of the other groups (*p* < 0.05) (Table 1). There were no significant differences in BMI between male and female participants in each group (*p* = 0.694).

### 3.2. Insulin Resistance-Markers of All Participants

As shown in Table 1, all patients had higher FBG, 2h-PPG, HbA1C, insulin and HOMA-IR values than the control healthy people (*p* < 0.05, *p* < 0.01, *p* < 0.001). Compared with that in all the disease groups, the level of FBG was significantly higher in the T2DM+C-HR group. The patients in the T2DM coexisting with hyperthyroidism had significantly higher levels of 2 h PPG than that in the T2DM coexisting with hypothyroidism (*p* < 0.001) (Table 1). Our results revealed that all patients with T2DM with thyroid dysfunctions had significantly higher insulin and HOMA-IR values than T2DM without thyroid dysfunctions (*p* < 0.01, *p* < 0.001). The highest significant values of insulin and HOMA-IR were observed in the T2DM coexisting with hyperthyroidism (*p* < 0.01, *p* < 0.001). It was found that patients with T2DM+SCTDs had a significantly lower percentage of HbA1C compared with T2DM with clinical thyroid dysfunctions (*p* < 0.01) (Table 1). There were no significant differences between male and female patients in insulin-resistance factors, including FBG (*p* = 0.492) insulin (*p* = 0.534) and HOMA-IR (*p* = 0.558), in each patient group. Our results revealed that T2DM patients with thyroid dysfunctions with BMI > 25 kg/m^2^ had higher HOMA-IR.values than those with BMI < 25 kg/m^2^ (Appendix A).

### 3.3. Thyroid-Hormone Profile in All Participants

As indicated in Table 2, there were statistically significant differences in the levels of fT3, fT4 and TSH between different groups. Patients with T2DM+C-HO showed the highest value of TSH compared with other disease groups (*p* < 0.0001), while patients with T2DM coexisting with hyperthyroidism had the lowest serum TSH concentration (*p* < 0.001, *p* < 0.0001). There were no significant differences in the serum levels of fT3 and fT4 between the T2DM+euthyroidism, the T2DM+SC-HO, and the control group. Patients in T2DM+C-HR had significantly higher serum concentrations of fT3 and fT4, compared with the T2DM+SC-HRr group (*p* < 0.001) (Table 2).The results for the thyroid autoantibody showed significantly higher anti-TPO in T2DM patients with overt/subclinical hypothyroidism (53.48% and 39.02%) when compared with T2DM patients with normal thyroid function (12.5%) and T2DM patients with overt/subclinical hyperthyroidism (21.21% and 13.88%, respectively). Our results revealed that sex and BMI had no obvious effects on differences in serum levels of thyroid hormones in all patient groups (*p* > 0.05).

### 3.4. Serum Adipocytokines in Different Studied Populations

As shown in Figure 1A–C, all disease groups had significantly higher serum concentrations of resistin, chemerin and visfatin, compared with the control, healthy people. For serum resistin-concentration, there was no significant difference between the T2DM patients and the T2DM+SCTDs. Elevated levels of serum resistin were observed in T2DM with clinical thyroid dysfunctions, compared with the T2DM patients (*p* < 0.00) (Figure 1A). Our results revealed that TD2M diabetic patients with SCTDs had the same serum chemerin values as the T2DM, euthyroid patients. Serum chemerin-concentrations were significantly higher in T2DM patients with clinical thyroid dysfunctions, compared with other disease groups (*p* < 0.001) (Figure 1B). As indicated in Figure 1C, hypothyroid patients coexisting with T2DM, showed elevated levels of serum visfatin, compared with other disease groups (*p* < 0.001). Serum concentration of visfatin in diabetic patients with hyperthyroidism showed no significant difference with that of patients in T2DM without thyroid dysfunction (Figure 1C). All groups showed no difference in serum resistin (*p* = 0.94), chemerin (*p* = 948) and visfatin (*p* = 0.529) concentrations between male and female patients. Our results revealed that T2DM patients with thyroid dysfunctions with BMI > 25 kg/m^2^ had higher serum resistin, chemerin and visfatin concentrations than those with BMI < 25 kg/m^2^ (Appendix A).

### 3.5. Proinflammatory Cytokines in All Participants

As presented in Figure 2A–C, all disease groups had significantly higher serum-concentrations of IL-6, IL1-β and CP, compared with the control, healthy people. The serum IL-6 level was higher in clinical hypo- and hyperthyroidism patients coexisting with T2DM, compared with other disease groups (*p* < 0.001, *p* < 0.0001) (Figure 2A). The T2DM patients coexisting with SCTDs showed no obvious change in serum concentration of IL-6 compared with T2DM without thyroid dysfunctions. The serum IL1-β concentration was higher only in the T2DM+C-HR patients (*p* < 0.001), while no significant differences were observed in serum IL1-β levels among other disease groups (Figure 2B). The patients in the T2DM+C-HO had a higher level of CRP than that in the other disease group (*p* < 0.01). No differences were observed in CRP levels between other disease groups (Figure 2C). Our results showed that sex had no significant impact on alterations of serum IL-6 (*p* = 0.871), IL1-β (*p* = 0.189) or CRP (*p* = 0.984), in all patient groups. BMI had a significant effect on changes of inflammatory factors in the serum of all patient groups. Our results revealed that patients with higher BMIs had higher levels of inflammatory cytokines in serum (*p* < 0.05) (Appendix A).

### 3.6. Association between Serum Adipocytokines and Insulin-Resistant Markers, Thyroid-Hormone Profile and Inflammatory Factors in T2DM Patients with Hypothyroidism

Our results revealed significant positive correlations between changes in visfatin, resistin, and chemerin concentration, and FBG concentration (*p* < 0.0001), while no such associations were observed between all adipocytikine concentrations and other insulin resistance-markers including insulin and HOMA-IR (Table 3). The serum concentrations of CRP, IL-6 and IL-1β showed significant positive correlations with serum concentrations of all three adipocytokines (*p* < 0.0001) (Table 3). The serum levels of visfatin, resistin and chemerin were negatively correlated with fT3 and fT4, and positively correlated with TSH serum-concentrations. BMI had significant positive associations with the serum levels of all adipocytokines (*p* < 0.05) (Table 3). The serum concentrations of visfatin and resistin showed positive associations with anti-TPO (*p* < 0.05) (Table 3).

### 3.7. Association between Adipocytokines and Metabolic and Inflammatory Factors in Hyperthyroid Patients with T2DM

As shown in Table 4, serum resistin—and chemerin concentrations were positively associated with insulin resistance-markers, including insulin, FBG and HOMA-IR (*p* < 0.0001). The visfatin concentration showed significant positive associations with FBG (*p* = 0.008) and HOMA-IR (*p* = 0.037), and had no association with serum level of insulin (*p* = 0.072). Regression analysis indicated the significant positive correlations between serum levels of chemerin and resistin and fT3 and fT4 (*p* < 0.0001), and significant negative association with TSH (*p* < 0.0001). The alteration of serum visfatin had no significant association with thyroid hormones (TSH; *p* = 0.068, fT3; *p* = 0.072, fT4; *p* = 0.086). Our results revealed that serum levels of resistin, chemerin and visfatin had significant positive associations with inflammatory factors including IL-6, IL-1β and CRP (*p* < 0.0001). Our results indicated that in T2DM patients with hyperthyroidism, serum concentrations of resistin and chemerin had no significant association with BMI and anti-TPO (*p* < 0.05).

## 4. Discussion

Studies have shown that diabetes and thyroid dysfunctions can be found to exist together where thyroid disease can affect glucose metabolism and the untreated thyroid dysfunctions can affect the management of diabetes [1,2]. Although the abnormal secretion of adipocytokines has been studied in patients with thyroid dysfunctions or T2DM, limited or controversial data are available about alterations of several adipocytokines in T2DM coexisting with subclinical thyroid dysfunctions. In the present study, alterations of serum visfatin, resistin and chemerin were studied in T2DM patients coexisting with clinical and subclinical thyroid dysfunctions, and their associations with thyroid hormones and inflammatory markers were evaluated.

HOMA-IR is a representative characteristic of diabetes and metabolic syndrome, and is associated with abnormal carbohydrate and lipid metabolism [28]. Our investigation indicated that T2DM patients with or without thyroid dysfunctions showed insulin resistance, which was characterized by hyperglycemia, hyperinsulinemia and an elevated level of HOMA-IR. It was also found that diabetic patients coexisting with thyroid dysfunctions had high levels of HbA1C, which was the most important determinant of long-term glycemic status in diabetic patients [29]. This suggests that poor glycemic-control as characterized by increased HbA1c is directly linked to the development of thyroid dysfunctions in T2DM patients. Our findings demonstrated that patients with clinical thyroid dysfunctions showed the highest insulin resistance-markers and HbA1C concentrations, compared with those with subclinical thyroid dysfunctions. In accordance with previous studies, our results revealed that hyper- and hypothyroidism was associated with insulin resistance [30,31]. The explanation to this apparent paradox may lie in the differential effects of thyroid hormones at the liver and peripheral-tissues level. In hyperthyroidism, impaired glucose tolerance may be the result of mainly hepatic insulin-resistance, whereas in hypothyroidism the available data suggests that the insulin resistance is associated with impaired insulin actions on peripheral tissues [1,2,6]. Our results indicated that diabetic patients with subclinical thyroid dysfunctions also showed insulin resistance, but to a lesser extent compared with clinical forms of thyroid dysfunctions. Controversial results were reported concerning the association of subclinical thyroid dysfunctions with insulin resistance. Subclinical hypo- or hyperthyroidism have been associated with insulin resistance in some, but not all, studies [30,31,32,33]. However, in some negative studies, hyperinsulinemia was reported in subclinical hypothyroid subjects, and interpreted as an early sign of impairment of glucose metabolism [34]. The heterogeneous nature of subclinical thyroid dysfunctions may partly explain this controversy. Because thyroid dysfunctions can induce or exacerbate cardiovascular disorders, insulin resistance should be evaluated and managed regularly, to reduce the impending risk of cardiovascular diseases in T2DM patients with thyroid dysfunctions [25].

In this study we found that serum concentrations of chemerin and resistin were higher in diabetic patients with or without thyroid dysfunctions, compared with the control group. T2DM patients with clinical hyper- or hypothyroidism had higher levels of chemerin and resistin, compared with the T2DM patients with subclinical thyroid dysfunctions. Our results also revealed that serum visfatin-concentration was higher in T2DM patients and in diabetic patients coexisting with clinical and subclinical hypothyroidism, compared with the control group. There were no obvious differences in the serum visfatin-levels between the T2DM patients and the T2DM patients with hyperthyroid dysfunction. Our correlation analysis indicated that in the T2DM patients coexisting with hyperthyroid dysfunctions, the elevated serum concentrations of chemerin and resistin had a positive association with changes in serum levels of T3 and T4. An inverse association was found between the serum concentrations of these adipocytokines with the serum concentrations of thyroid hormones in the T2DM patients coexisting with hypothyroid dysfunctions. In the T2DM patients coexisting with hypothyroidism, the serum chemerin- and resistin-concentrations had a positive correlation with FBG only, while in the T2DM patients with hyperthyroid dysfunction, chemerin and resistin showed positive association with FBG and HOMA-IR. Taken together it was concluded that in the T2DM patients with hyperthyroidism, insulin resistance and elevated levels of thyroid hormones may have decisive role in the secretion of chemerin and resistin into the blood, while in the T2DM patients with hypothyroidism, the elevation of these adipocytokines was relatively independent of thyroid hormones and insulin action. An inverse pattern was observed between the alteration of serum visfatin and thyroid hormones. The decrease in thyroid hormones was associated with the increase in the visfatin hormone, and the serum changes of this hormone were relatively independent of insulin-resistance factors. These findings showed that chemerin, visfatin and resistin might exert different biological actions, depending on the thyroid-hormone levels and insulin-resistance condition in the T2DM patients coexisting with thyroid dysfunctions. To support these findings, contradictory results have been reported on the role of these adipocytokines in whole-body metabolism and insulin resistance. Previous reports indicated that plasma chemerin-levels are increased in diet-induced obese mice, and reduced by overnight fasting. In contrast, there was no change in plasma chemerin-levels in NMRI mice on a high-fat or cafeteria diet [35]. It has been found that visfatin causes an increase in insulin secretion, while it has also been reported that elevated serum visfatin-levels are linked to β-cell dysfunction [36,37]. These findings demonstrated that physiological and supraphysiological concentrations, or the exposure-time of cells to these adipocytokines, may have different consequences in pathological conditions. Thus, it is unclear whether raised chemerin-, resistin- and visfatin-levels in T2DM patients with thyroid dysfunctions promote or improve metabolic dysfunctions. Further research is needed to provide a definite answer.

To the best of our knowledge, this is the first study to investigate the alterations of resistin, chemerin and visfatin in the serum of T2DM patients coexisting with subclinical thyroid dysfunctions. Controversial results were reported concerning the alterations of serum concentrations of adipocytokines in thyroid dysfunctions or T2DM. Edrees et al. reported that the serum chemerin-level in the animal model of thyroid dysfunctions increases in hyperthyroid rats and decreases in the hypothyroid condition in comparison with the control group [11]. They also reported that serum levels of chemerin have a negative correlation with the serum level of T3 and T4, and a positive correlation with the serum level of TSH, in all studied groups [11]. There are few clinical studies that have examined changes in chemerin in subclinical thyroid dysfunctions. The limited data indicates that the chemerin level is comparable between patients with subclinical hypothyroidism and the control groups, but levels of this adipocytokine significantly increase in hypothyroid patients. Recently, in an experimental study of subclinical hypothyroidism, a higher level of chemerin compared with that of control rats has been reported [11,38].

The role of thyroid hormones in the regulation of visfatin concentrations is controversial. The higher plasma visfatin-concentration in hyperthyroid patients compared with controls, has previously been reported. Chu et al. [17] found that visfatin concentrations were higher in patients with hyperthyroidism, compared with controls. Alshaikh et al. [10] reported lower serum visfatin-levels in hyperthyroid patients than in controls. A similar observation was reported by Ozkaya et al. [19], who found that patients with hyperthyroidism had lower concentrations of visfatin than the control group.

Data related to alterations of resistin in thyroid dysfunctions is very limited. Nogueiras et al. [39] reported that resistin mRNA levels increase in adipose tissue in hypothyroid rats, whereas they are severely reduced in hyperthyroid rats. Conflicting results are also available concerning the elevation of plasma resistin-levels in hypothyroid patients [12].Some previous reports have shown that TSH receptors are expressed in adipose tissues, and TSH can regulate the expression of several adipose-derived hormones. Therefore, changes in the thyroid hormones and TSH may affect the release of adipocytokines, so there is a possible relationship between thyroid status, thyroid dysfunction and adipocytokines [40]. One important finding of our study was that patients with T2DM coexisting with subclinical thyroid dysfunctions and with normal thyroid-hormone values showed no significant differences in the levels of all studied adipocytokines, compared with those in the T2DM patients without thyroid dysfunctions. As described in our results, serum concentrations of thyroid hormones in the T2DM patients with subclinical thyroid dysfunctions were within the reference range, while TSH concentrations showed abnormal levels. These findings indicated that thyroid hormones and TSH may play a role in regulating adipocytokine secretion in clinical thyroid dysfunctions whereas only thyroid hormones seem to influence the serum concentrations of adipocytokines in subclinical thyroid dysfunctions.

The increased concentrations of studied adipocytokine in the T2DM patients coexisting with thyroid dysfunctions, regardless of the fluctuations of thyroid hormones, may be attributed to hyperinsulinemia or hyperglycemia. To confirm this hypothesis, previous reports indicated that insulin could stimulate the secretion of visfatin and chemerin from adipose tissue [41,42]. The increase in serum adipocytokines in the T2DM patients with clinical thyroid dysfunctions may have dual protective and harmful effects under an insulin-resistance condition. Confirming this opinion, previous findings have shown that visfatin stimulates glucose uptake in adipocytes and myocytes by the activation of insulin signaling pathways [43]. In contrast, the elevation of chemerin and resistin in our study may have an adverse effect on the progression of insulin resistance in diabetic patients with thyroid dysfunction. This opinion may be supported by previous findings indicating that chemerin and resistin induce insulin resistance in skeletal-muscle cells at the level of insulin-receptor glucose uptake [44].

Although the exact mechanism mediating the elevation of these adipokines in the T2DM patients suffering from thyroid dysfunctions remains obscure, one possible explanation for these findings is that increased pro-inflammatory cytokine levels and the consequent inflammatory milieu can affect adipocytokine levels. Our results showed the elevated levels of inflammatory markers such as IL1-β, IL-6 and CRP in T2DM patients suffering from thyroid dysfunctions, compared with healthy people. We also observed the positive association between alterations in serum levels of resistin, visfatin and chemerin, and changes in the serum concentration of IL1-β, IL-6 and CRP levels. These findings indicated that visfatin, resistin and chemerin may affect proinflammatory-cytokine production in T2DM patients with thyroid dysfunctions. A previous report has shown that chemerin promotes inflammation through the ChemR23-receptor activation, by acting on the chemotactic mechanisms of immune cells and the secretion of proinflammatory cytokines including IL-6, TNF-α and IL-1β [45]. It has been also found that visfatin stimulates monocytes to produce proinflammatory cytokines such as IL-1, IL-6, IL-8 and TNF-α, and induces the chemotaxis of activated mononuclears [46]. The binding of resistin to CAP-1, as a possible receptor, also stimulates the expression of proinflammatory cytokines by activating the NF-κβ signaling pathways [47]. Taken together, it plausible that high serum levels of these adipocytokines may enhance proinflammatory cytokine secretion in a positive feedback loop, which in turn has a detrimental role in thyroid dysfunction-related pathologies in T2DM patients.

Our results revealed that patients with BMI > 25 kg/m^2^ had higher concentrations of serum levels of resistin, chemerin and visfatin, insulin resistance-markers and serum levels of inflammatory cytokines, compared with patients with BMI > 25 kg/m^2^. We also found a significant positive relationship between BMI value and serum adipocytokines, inflammatory factors and HOMA-IR in T2DM patients with hypothyroidism. We did not observe such an association in diabetic patients with hyperthyroidism. This finding indicates that in diabetic patients with hypothyroidism, obesity may be an important factor affecting the secretion of adipose-derived inflammatory factors and the progression of insulin resistance. In line with our study, several previous studies indicated that obesity was correlated with an increased risk of hypothyroidism, and patients with BMI ≥ 28 kg/m^2^ showed an increased risk of overt hypothyroidism. However, no significant association has been observed between obesity and an increased risk of hyperthyroidism in T2DM patients [48,49]. Several mechanisms may be proposed to explain the bidirectional association between high BMI and thyroid diseases in T2DM patients. In patients with high BMI, over-loading the adipose tissue results in chronic low-grade inflammation and increased secretion of proinflammatory cytokines and inflammatory adipocytokines [50,51]. The increased production of inflammatory cytokines may inhibit the thyroid cell functions and induce thyroid diseases. Confirming this opinion, previous findings indicated that inflammatory cytokines such as TNF-α and IL-1β influence the iodide-uptake activity of human thyroid cells by inhibiting the expression of the sodium/iodide symporter [52]. The reduction of thyroid-hormone production in patients with high BMI and insufficient thyroid-hormone production may reduce the adipose-tissue metabolic rate, resulting in profound insulin resistance and more obesity. Taken together, these findings suggest that BMI may be a decisive factor for the development of hypothyroidism in T2DM patients.

The analysis of the anti-TPO antibody revealed the positive association between the anti-TPO value and the serum concentrations of visfatin and resistin in the T2DM patients with hypothyroidism. We also found that the incidence of anti-TPO antibody was significantly higher in hypothyroid subjects with BMI > 25 kg/m^2^ than in hypothyroid patients with a BMI < 25 of kg/m^2^. A previous study has speculated that obesity itself may be a risk factor for thyroid autoimmunity [53]. It has been shown that various adipokines secreted by the adipose tissue such as resistin, may stimulate immune responses by activating several inflammatory signaling pathways [54]. Moreover, chronic inflammation in obesity may act as a trigger for autoimmunity. Taken together it was concluded that elevation of visfatin and resistin in T2DM may lead to impaired thyroid function by inducing thyroid-autoimmunity responses.

## 5. Conclusions

Collectively, our data showed that serum concentrations of visfatin, chemerin, and resistin increased in the T2DM patients suffering from clinical and subclinical thyroid dysfunctions. Similar results were noted in the case of T2DM coexisting with subclinical thyroid dysfunctions, but to a lesser extent. Furthermore, our observations suggest that the dysregulated secretion of thyroid hormones concomitant with the elevation of adipocytokines and inflammatory markers may be an important mechanism of thyroid dysfunction pathologies in T2DM patients. We also found that BMI was an important factor for the incidence of hypothyroid dysfunction in T2DM patients, indicating an interplay of excess weight in adult T2DM pathophysiology. Further studies are needed to clarify whether the assessment of the alteration of the adipocytokine pattern, together with other inflammatory factors, can be considered as reliable biomarkers which help to elucidate the different clinical features of T2DM coexisting with thyroid dysfunctions.

## Figures and Tables

**Figure 1 biomedicines-11-00346-f001:**
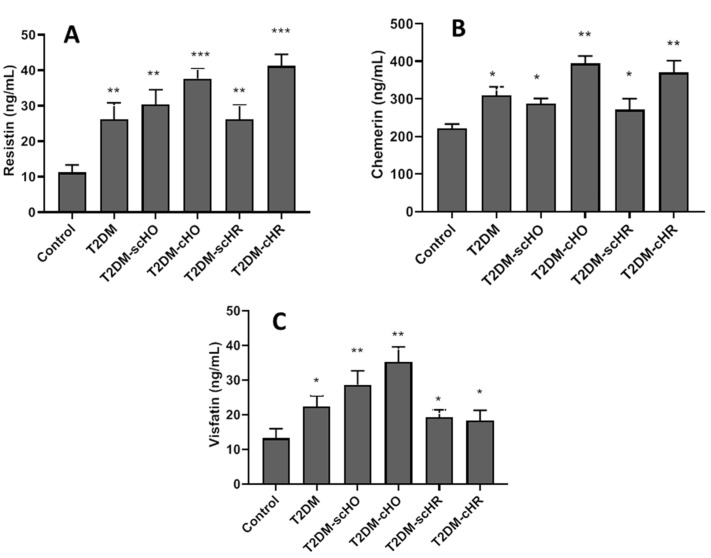
Serum levels of resistin (**A**), chemerin (**B**) and visfatin (**C**) in different studied populations. Data are means ± SD, and *, ** and *** indicate significant differences between all patient groups with healthy populations at *p* < 0.05, *p* < 0.01 and *p* < 0.001, respectively. T2DM: type 2 diabetes mellitus; SC-HO: subclinical hypothyroidism; C-HO: clinical hypothyroidism; SC-HR: subclinical hyperthyroidism; C-HR: clinical hyperthyroidism.

**Figure 2 biomedicines-11-00346-f002:**
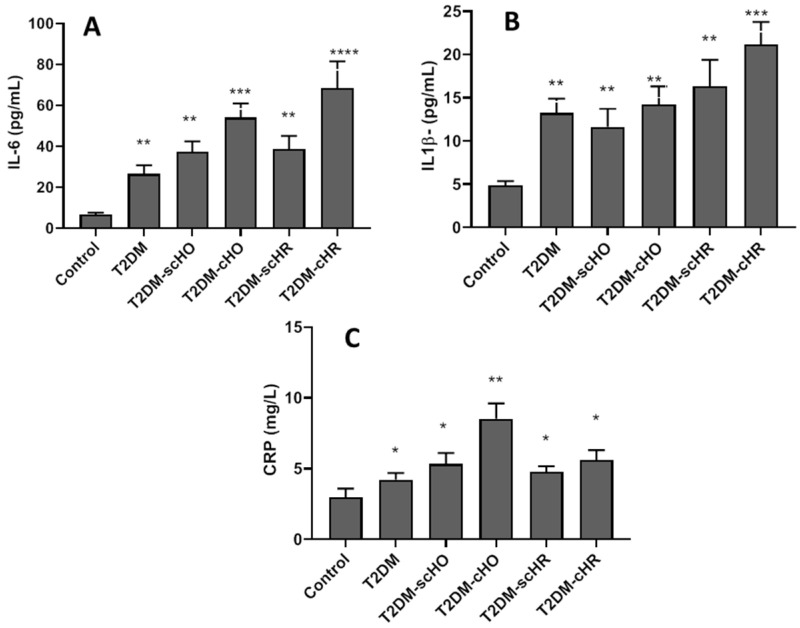
Serum levels of IL-6 (**A**), IL1-β (**B**) and CRP (**C**) in different studied populations. Data are means ± SD, and *, **, *** and **** indicate significant differences between all patient groups with healthy populations at *p* < 0.05, *p* < 0.01, *p* < 0.001 and *p* < 0.0001, respectively. T2DM: type 2 diabetes mellitus; SC-HO: subclinical hypothyroidism; C-HO: clinical hypothyroidism; SC-HR: subclinical hyperthyroidism; C-HR: clinical hyperthyroidism.

**Table 1 biomedicines-11-00346-t001:** Clinical and demographic data of studied population.

Variable (Mean ± SD)	Healthy	T2DM Euthyroidism	T2DM SC-Hypo	T2DM C-Hypo	T2DM SC-Hyper	T2DM C-Hyper
N	55	64	41	43	36	33
Gender (F/M)	27/28	36/31	24/17	27/16	17/19	14/19
Age (years)	51.6 ± 10.4	57.6 ± 9.8	53.5 ± 11.2	53.5 ± 11.2	48.5 ± 8.6	50.3 ± 6.7
BMI (kg/m^2^)	21.2 ± 1.6	28.6 ± 2.2 *	28.9 ± 1.7 *	30.4 ± 2.8 *	25.3 ± 3.1	23.7 ± 2.2
FBG (mg/dL)	87.2 ± 11.6	138.8 ± 23.2 *	141.70 ± 16.5 *	150.4 ± 19.3 *	150.3 ± 16.1 *	168.5 ± 13.8 **
2h PPG (mg/dL)	95.4 ± 6.3	211.7 ± 20.3 **	218.9 ± 13.8 **	224.6 ± 19.6 **	253.0 ± 20.2 ***	267.8 ± 11.8 ***
Insulin (uU/mL)	5.3 ± 1.4	8.2 ± 3.1 *	11.4 ± 2.2 **	12.2 ± 3.6 **	17.6 ± 4.2 ***	20.2 ± 6.2 ***
HOMA-IR (mM*mU/L)	1.1 ± 0.3	6.4 ± 0.6 **	5.8 ± 0.8 **	6.1 ± 0.6 **	5.6 ± 0.5 **	7 ± 0.7 ***
HbA1c (%)	4.7 ± 1.1	6.7 ± 1.8 *	6.5 ± 1.3 *	7.8 ± 0.7 **	6.3 ± 0.9 *	7.7 ± 1.2 **

Data are presented as mean ± standard deviation, SD; T2DM: type 2 diabetes mellitus; SC-HO: subclinical hypothyroidism; C-HO: clinical hypothyroidism; SC-HR: subclinical hyperthyroidism; C-HR: clinical hyperthyroidism; BMI: body mass index; FBG: fasting blood glucose; 2h-PPG: 2 h postprandial glucose; HOMA-IR: homeostatic-model assessment for insulin resistance; *, ** and *** indicate significant differences between all patient groups with healthy populations at *p* < 0.05, *p* < 0.01 and *p* < 0.001, respectively.

**Table 2 biomedicines-11-00346-t002:** Thyroid-hormones profiles of studied population. Data are presented as mean ± standard deviation, SD; T2DM: type 2 diabetes mellitus; SC-HO: subclinical hypothyroidism; C-HO: clinical hypothyroidism; SC-HR: subclinical hyperthyroidism; C-HR: clinical hyperthyroidism; TSH: thyroid-stimulating hormone; FT3: free T3, FT4: free T4; Anti-TPO: anti-thyroid peroxidase antibody; *, **, *** and **** indicate significant differences between all patient groups with healthy populations at *p* < 0.05, *p* < 0.01, *p* < 0.001 and *p* < 0.0001, respectively.

Variable (Mean ± SD)	Healthy	T2DM Euthyroidism	T2DM SC-HO	T2DM C-HO	T2DM SC-HR	T2DM C-HR
N	55	64	41	43	36	33
TSH (μIU/mL)	2.24 ± 0.49	1.87 ± 0.36	11.31 ± 2.46 **	63.27 ± 8.19 ****	0.16 ± 0.047 ***	0.04 ± 0.01 ****
FT3 (pmol/L)	4.21 ± 0.61	3.81 ± 0.52	3.81 ± 0.46	2.43 ± 0.72 *	4.82 ± 0.71 **	11.49 ± 2.48 ***
FT4 (pmol/L)	10.42 ± 1.8	9.84 ± 2.06	8.78 ± 1.21	4.22 ± 0.71 *	12.32 ± 0.97 **	30.72 ± 4.67 ***
TPO-Ab positive; *n* (%)	4 (7.27)	8 (12.5)	16 (39.02) ***	23 (53.48) ****	5 (13.88)	7 (21.21) **

**Table 3 biomedicines-11-00346-t003:** Association between serum levels of adipocytokines and insulin-resistant markers, thyroid-hormone profile and inflammatory factors in T2DM patients coexisting with hypothyroidism.

T2DM with Clinical and Subclinical Hypothyroidism						
Variables	Visfatin		Resistin		Chemerin	
	*β*(*r*)	*p*	*β*(*r*)	*p* < 0.0001	*β*(*r*)	*p*
TSH	0.861	*p* < 0.0001	0.829	*p* < 0.0001	0.988	*p* < 0.0001
fT3	−0.751	*p* < 0.0001	−0.767	*p* < 0.0001	−0.902	*p* < 0.0001
fT4	−0.868	*p* < 0.0001	−0.838	*p* < 0.0001	−0.983	*p* < 0.0001
Anti-TPO	0.362	0.039	0.466	0.002	0.226	0.078
CRP	0.810	*p* < 0.0001	0.722	*p* < 0.0001	0.820	*p* < 0.0001
IL-6	0.811	0.075	0.827	*p* < 0.0001	0.954	*p* < 0.0001
IL1-β	0.725	0.018	0.705	0.001	0.737	*p* < 0.0001
FBG	0.726	*p* < 0.0001	0.650	*p* < 0.0001	0.795	*p* < 0.0001
HOMA-IR	0.141	0.267	0.112	0.374	0.171	0.278
Insulin	0.275	0.098	0.124	0.302	0.174	0.269
Visfatin	1	*p* < 0.0001	0.262	0.157	0.262	0.157
Resistin	0.262	0.157	1	*p* < 0.0001	0.898	*p* < 0.0001
Chemerin	0.378	0.096	0.898	*p* < 0.0001	1	*p* < 0.0001
BMI	0.378	0.019	0.334	0.031	0.447	0.003

CRP: C-reactive protein; FBG: fasting blood glucose; HOMA-IR: homeostatic-model assessment for insulin resistance; *β*(*r*): linear regression coefficient.

**Table 4 biomedicines-11-00346-t004:** Association between serum levels of adipocytokines and insulin-resistant markers, thyroid-hormone profile and inflammatory factors in T2DM patients coexisting with hyperthyroidism.

T2DM with Clinical and Subclinical Hyperthyroidism						
Variables	Visfatin		Resistin		Chemerin	
	*β*(*r*)	*p*	*β*(*r*)	*p* < 0.0001	*β*(*r*)	*p*
TSH	−0.321	0.068	−0.910	*p* < 0.0001	−0.975	*p* < 0.0001
fT3	0.317	0.072	0.906	*p* < 0.0001	0.982	*p* < 0.0001
fT4	0.304	0.086	0.902	*p* < 0.0001	0.864	*p* < 0.0001
Anti-TPO	0.202	0.078	0.266	0.137	0.165	0.351
CRP	0.164	0.361	0.720	*p* < 0.0001	0.722	*p* < 0.0001
IL-6	0.315	0.075	0.880	*p* < 0.0001	0.958	*p* < 0.0001
IL1-β	0.409	0.018	0.569	0.001	0.668	*p* < 0.0001
FBG	0.451	0.008	0.686	*p* < 0.0001	0.787	*p* < 0.0001
HOMA-IR	0.365	0.037	0.793	*p* < 0.0001	0.759	*p* < 0.0001
Insulin	0.318	0.072	0.783	*p* < 0.0001	0.849	*p* < 0.0001
Visfatin	1	*p* < 0.0001	0.267	0.122	0.319	0.069
Resistin	0.267	0.122	1	*p* < 0.0001	0.898	*p* < 0.0001
Chemerin	0.319	0.069	0.898	*p* < 0.0001	1	*p* < 0.0001
BMI	−0.137	0.213	−0.221	0.089	−0.197	0.137

CRP: C-reactive protein; FBG: fasting blood glucose; HOMA-IR: homeostatic-model assessment for insulin resistance; *β*(*r*): linear regression coefficient.

## Data Availability

Derived data supporting the findings of this study are available from the corresponding author on request.

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
