# Peer review of "Type 2 Diabetes Mellitus Coincident with Clinical and Subclinical Thyroid Dysfunctions Results in Dysregulation of Circulating Chemerin, Resistin and Visfatin"

_biomedicines, 2023, doi:10.3390/biomedicines11020346_

Round 1
Reviewer 1 Report
Adipocytes have a primary role in inducing insulin resistance. The secretion of adipocytokines is an area of major interest. The authors have performed a clinical study comparing both adipocytokine levels and those of inflammatory markers in patients with diabetes and patients with diabetes and thyroid abnormalities. The thyroid abnormalities have been defined as clinical hypothyroidism, subclinical hypothyroidism, subclinical hyperthyroidism, and clinical hyperthyroidism. The results are fascinating in showing substantial differences in adipocytokine and inflammatory marker levels compared to a relatively small control population. What is lacking is an assessment of thyroid autoantibodies, including thyroid stimulating antibody, thyroid inhibitory immunoglobulin, and antithyroid peroxidase in their population. That is an important limitation in interpretation of their results. It would be extremely valuable to present comparison data in cohorts with thyroid disease and no diabetes. Nonetheless, as it stands, this is a valuable pilot study.
It is really necessary to explain the characteristics of thyroid disease in Iran.
In particular, their ability to recruit these large cohorts, with almost equal sex distribution, suggests that thyroid disease may be more common in Iran than in Western countries.
SPECIFIC COMMENTS
1) They have a large population of patients with diabetes and hyperthyroidism. Can they explain how they were able to recruit such a large cohort?
2) Thyroid disease is much more common in women than men. Yet their distribution of patients is fairly equivalent between the sexes.
Author Response
November , 11, 2022
Dear Editor-in-Chief of Biomedicines
Coauthors and I very much appreciated the encouraging, critical and constructive comments on our manuscript "Type 2 diabetes mellitus coincident with clinical and subclinical thyroid dysfunctions results in dysregulation of circulating chemerin, resistin and visfatin" by the reviewers and are grateful for the insightful comments on and valuable improvements to our paper. The authors have carefully considered the comments and tried our best to address every one of them. We hope the manuscript after careful revisions meet your high standards. All corrections were highlighted in the text. Below we provide the point-by-point responses.
Response to reviewer1
Thank you so much for your valuable suggestions and comments in detail, which were very helpful for improving the manuscript
-The TPOAb data were added to the text as correctly by the reviewer.
-The status of thyroid disease in diabetic patients in Iranian adults was added to the text as advised by the reviewer.
-The purpose of the present study was not to investigate the prevalence of T2DM coincident with thyroid dysfunctions in Iran. The samples of the present study were collected between 2016-2020 from participants referred to 9 branches of our laboratories in Khuzestan provinces. A large number of serum samples were collected during the annual screening of health status of workers and employees of the oil and gas, and steel industries located in the Khuzestan province. As indicated by the reviewer, the prevalence of thyroid diseases is more common in female than male, but because most of the available samples in the present study were taken from men, the distribution of patients is relatively equivalent between the sexes. As a result, our study was not a cohort study and the minimum number of samples that met the inclusion criteria was selected for entering the study.
In accordance with our research, many previous published researches have used similar control sample size to determine the fluctuation of serum adipocytokines in serum of patients with different diseases. We address the following articles to confirm our opinion:
-Sawicka-Gutaj N, Zybek-Kocik A, Klimowicz A, Kloska M, Mańkowska-Wierzbicka D, Sowiński J, Ruchała M. Circulating visfatin in hypothyroidism is associated with free thyroid hormones and antithyroperoxidase antibodies. International journal of endocrinology. 2016 Jan 17;2016. (39 control sample).
-Eke Koyuncu C, Turkmen Yildirmak S, Temizel M, Ozpacaci T, Gunel P, Cakmak M, Ozbanazi YG. Serum resistin and insulin-like growth factor-1 levels in patients with hypothyroidism and hyperthyroidism. Journal of thyroid research. 2013;2013.
-Yasar HY, Demirpence M, Colak A, Yurdakul L, Zeytinli M, Turkon H, Ekinci F, Günaslan A, Yasar E. Serum irisin and apelin levels and markers of atherosclerosis in patients with subclinical hypothyroidism. Archives of endocrinology and metabolism. 2019;63:16-21.
We hope that the respected reviewers find that their concerns have been adequately addressed in the revised article, and this could meet the requirements for publication in the Biomedicines.
Reviewer 2 Report
The authors present a cross-sectional analysis on the impact of clinical and subclinical thyroid disorder on adipokine levels and metabolism in patients with T2DM.
The overall rationale of the paper is clear.
Methods: Presented clearly, conducted mainly properly. Comparisons should be adjusted for sex, as HbA1c, but also adipokines are affected by this confounder. BMI also serves as major confounder and should be included in ANOVA models.
The correlation analyses (Table 3.6. and 3.7.) should be adapted into adjusted linear regression analyses, which include sex and BMI as co-variates.
Results: In general plausible, but currently confounded by sex and BMI.
Discussion: Will be evaluated after revision.
Minor: Please replace FBS by FPG.
Author Response
November , 11, 2022
Dear Editor-in-Chief of Biomedicines
Coauthors and I very much appreciated the encouraging, critical and constructive comments on our manuscript "Type 2 diabetes mellitus coincident with clinical and subclinical thyroid dysfunctions results in dysregulation of circulating chemerin, resistin and visfatin" by the reviewers and are grateful for the insightful comments on and valuable improvements to our paper. The authors have carefully considered the comments and tried our best to address every one of them. We hope the manuscript after careful revisions meet your high standards. All corrections were highlighted in the text. Below we provide the point-by-point responses.
Response to reviewer 2
Thank you so much for your valuable suggestions and comments in detail, which were very helpful for improving the manuscript
-The linear regression analysis with sex and BMI as covariates was performed using SPSS software to determine the associations between serum adipocytokineas and other factors. Sex and BMI was included in one way ANOVA analysis.The method, results and discussion sections were revised based on final statistical analysis.
-FBS was replaced with FBG as advised by the reviewer
We hope that the respected reviewers find that their concerns have been adequately addressed in the revised article, and this could meet the requirements for publication in the Biomedicines.
Round 2
Reviewer 1 Report
This is a valuable pilot study showing differences in adipocytokine level in patients with clinical and subclinical hypo and hyperthyroidism. The findings relative to antiTPO levels are particularly interesting. It is troubling that they did not present regression values for antiTPO. Also I can’t find regression values for BMI.
SPECIFIC COMMENTS
Previous reports have indicated that the 39 prevalence of overt and subclinical hypothyroidism in patients with T2DM in Iran is between 40 8-20%, respectively. The prevalence of overt and subclinical hyperthyroidism in general popula-41 tion with T2DM in Iran has reported between 1-2 %. Thyroid
HAS BEEN REPORTED
Hyperthyroidism is frequently developed with
IS ASSOCIATED WITH
Thyroid stimulating hormone (TSH) 58 induces secretion of adipokines from adipose tissue and preadipocytes by signaling to the TSH 59 receptor (TSHR) protein [13].
MORE ON WHITE OR BROWN FAT OR THE SAME? REFERENCES
adipose derived adipocytokines that 66 changes in their serum concentrations have been reported previously in diabetes
CHANGES IN SERUM CONCENTRATION OF… HAVE BEEN PREVIOUSLY REPORTED IN…
in hyperthyroid patients [17-18], while another report lower visfatin concentration in 76 hyperthyroid patients
OTHERS REORT LOWER CONCENTRATIONS IN HYPERTHYROIDISM….
The diagnosis of T2DM was performed accordance with the 2014 American Diabetes 125 Association diagnostic criteria for diabetes
PERFORMED IN ACCORDANCE
3.4. Serum adipocytokines in different studied population 286
As shown in Fig 1A-C, all disease groups had significant higher serum concentrations of resistin, 287 chemerin and visfatin compared with control, healthy people. Serum resistin concentration had no 288 significant difference between T2DM patients and T2DM+SCTDs. Elevated levels of serum re-289 sistin were observed in T2DM with clinical thyroid dysfunctions compared with T2DM patients (p 290 < 0.00) (Fig 1A). Our results reveled that TD2M diabetic patients with SCTDs had the same serum 291 chemerin values as the T2DM, euthyroid patients. Serum chemerin concentrations were signifi-292 cantly higher in T2DM patients with clinical thyroid dysfunctions compared with other disease 293 groups (p <0.001) (Fig 1B). As indicated in Fig 1C, hypothyroid patients coexisted with T2DM, 294 showed elevated levels of serum visfatin compared with other disease groups (p <0.001). 295 Serum concentration of visfatin in diabetic patients with hyperthyroidism had no signif-296 icant difference with that in T2DM without thyroid dysfunction (Fig 1C). All groups did 297 not differ in serum resistin (p =0.94), chemerin (p = 948) and visfatin (p = 0.529) con-298 centrations between male and female patients. GLM analysis indicated that BMI had 299 profound effect on serum levels of resistin, chmerin and visfatin in all patients groups (p < 300 0.05). Serum levels of these adipocytokines were significantly higher in the high BMI 301 participants than in the low BMI participants (p < 0.05).
THIS WHOLE PARAGRAPH IS HARD TO FOLLOW. THE GRAPHS ARE FINE. PERHAPS THE AUTHORS CAN DO A SUMMARY TABLE SHOWING WHICH ADIPOCYTOKINES ARE DIFFERENT IN THEIR COHORTS
Our results revealed that, BMI had profound effect on serum levels of resistin, chemerin and 597 visfatin, insulin resistance markers and serum levels of inflammatory cytokines in T2DM patients 598 coexisted with thyroid dysfunctions, in particular hypothyroid patients
WHERE IS THE BMI ANALYIS? DOES NOT APPEAR IN THE TABLESg
Author Response
December, 02, 2022
Dear Editor-in-Chief of Biomedicines
Coauthors and I very much appreciated the encouraging, critical and constructive comments on our manuscript "Type 2 diabetes mellitus coincident with clinical and subclinical thyroid dysfunctions results in dysregulation of circulating chemerin, resistin and visfatin" by the reviewers and are grateful for the insightful comments on and valuable improvements to our paper. The authors have carefully considered the comments and tried our best to address every one of them. We hope the manuscript after careful revisions meet your high standards. All corrections were highlighted in the text. Below we provide the point-by-point responses.
Response to reviewer 2
Thank you so much for your valuable suggestions and comments in detail, which were very helpful for improving the manuscript
- The regression values for BMI and anti-TPO were added to the text and discussed in discussion section.
-All grammatical and writing errors were edited through the text as advised by the reviewer.
-We think that adding another table for presentation of serum levels of adipocytokines (section3.4) will duplicate the results presented in the graphs. As a result, we request the reviewer allow us to keep the section 3.4 as the current format.
-In the present study T2DM patients with thyroid dysfunctions were divided into two groups based on BMI, and serum adipocytokines concentrations, inflammatory factors values and HOMI-IR were compared between two groups. We added this data in the text as supplementary table 1 for further clarification of results.
We hope that the respected reviewer find that their concerns have been adequately addressed in the revised article, and this could meet the requirements for publication in the Biomedicines.

Reviewer 2 Report
The authors have revised their manuscript in accordance to the reviewer's suggestions. Thanks a lot for that.
A very minor point is new: Line 240: "changes" --> "differences"; as your analysis is cross-sectional.
I also suggest to reduce decimals in accordance to raw data precision, e.g BMI, HbA1c and glucose levels with just one decimal.
Author Response
September, 02, 2022
Dear Editor-in-Chief of Biomedicines
Coauthors and I very much appreciated the encouraging, critical and constructive comments on our manuscript "Type 2 diabetes mellitus coincident with clinical and subclinical thyroid dysfunctions results in dysregulation of circulating chemerin, resistin and visfatin" by the reviewers and are grateful for the insightful comments on and valuable improvements to our paper. The authors have carefully considered the comments and tried our best to address every one of them. We hope the manuscript after careful revisions meet your high standards. All corrections were highlighted in the text. Below we provide the point-by-point responses.
Response to reviewer1
Thank you so much for your suggestions and comments in detail, which were very helpful for improving the manuscript
-The “changes” was replaced in line 240.
-The data in table 1 was presented by one decimal.

Round 3
Reviewer 1 Report
Excellent study showing that adipocytokine levels are abnormal in type 2 diabetes. In patients with clinically abnormal hypo or hyperthyroidism (showing abnormal T3 and T4), there is a further abnormality, beyond that seen in type 2 diabetes with subclinical hypo or hyperthyroidism (only TSH abnormal). The presentation must be reorganized to clearly explain these fascinating findings. Needs a regression analysis for BMI and for T4, T3 as well.
SPECIFIC COMMENTS
Our results revealed that clinical thyroid dysfunction in T2DM patients was associated with elevation of circulating resistin, chemerin, visfatin and inflammatory factors, while, no such alteration was detected in T2DM coincident with subclinical thyroid dysfunction.
WHAT YOU ARE TRYING TO POINT OUT IS THAT TYPE 2 DIABETES PATIENTS WITH NORMAL TSH DID NOT DIFFER IN ADIPOCYTOKINE LEVELS FROM TYPE 2 DIABETES PATIENTS WITH ABNORMAL TSH (SUBCLINICAL)? THEN FIGURE 1 NEEDS TO HAVE TWO DIFFERENT SYMBOLS FOR PROBABLILITY SIGNIFICANCE, ONE FOR DIFFERENCE FROM NORMAL WHICH IS ALL THE TYPE 2 DIABETES CATEGORIES AND A DIFFERENT SYMBOL, PERHAPS ! TO SHOW THE DIFFERENCE OF THE ABNORMAL FROM THE NORMAL TSH PATIENTS
Previous reports has been reported
ELIMINATE START WITH: THE PREVALENCE OF…
Previous study has shown
PRIOR STUDIES HAVE SHOWN
while other report
WHILE OTHERS REPORT
Author Response
We would like to thank the reviewers for their detailed feedback to improve our submission. We have considered each comment carefully and revised our manuscript to address the issues raised. Below we provide the point-by-point responses.
Response to reviewer
-The regression analysis data for BMI, fT3 and fT4 have been added in previous version of manuscript and corresponding data are presented in tables 3 and 4.We think that the presented results regarding the regression analysis of BMI and thyroid hormones are adequately performed and presented in the text.
-Although I agree with the opinion of the respected referee regarding the lack of difference in the amount of adipocytokines between diabetic patients with subclinical thyroid disorders and those without subclinical thyroid dysfunctions, but the comparison of groups in Fig 1 is not based on TSH levels and
using different symbols to show this difference in Fig 1 is not appropriate. We preferred to discuss these findings in the discussion section (paragraph 6). -All grammatical errors were edited through the text as advised by the reviewer.